# Dual-Responsive Starch Hydrogels via Physicochemical Crosslinking for Wearable Pressure and Ultra-Sensitive Humidity Sensing

**DOI:** 10.3390/s25165006

**Published:** 2025-08-13

**Authors:** Zi Li, Jinhui Zhu, Zixuan Wang, Hao Hu, Tian Zhang

**Affiliations:** 1Electronic Information School, Wuhan University, Wuhan 430072, China; 2022302121441@whu.edu.cn (Z.L.); zhujinhui@whu.edu.cn (J.Z.); 2023302121288@whu.edu.cn (Z.W.); 2Suzhou Institute of Wuhan University, Suzhou 215000, China

**Keywords:** starch, dual-network architecture, wearable sensor, dual-responsive

## Abstract

Flexible hydrogel sensors demonstrate emerging applications, such as wearable electronics, soft robots, and humidity smart devices, but their further application is limited due to their single-responsive behavior and unstable, low-sensitivity signal output. This study develops a dual-responsive starch-based conductive hydrogel via a facile “one-pot” strategy, achieving mechanically robust pressure sensing and ultra-sensitive humidity detection. The starch-Poly (2,3-dihydrothieno-1,4-dioxin)-poly (styrenesulfonate) (PEDOT:PSS)-glutaraldehyde (SPG) hydrogel integrates physical entanglement and covalent crosslinking to form a porous dual-network architecture, exhibiting high compressive fracture stress (266 kPa), and stable electromechanical sensitivity (ΔI/I_0_, ~2.3) with rapid response (0.1 s). In its dried state (D-SPG), the film leverages the starch’s hygroscopicity for humidity sensing, detecting minute moisture changes (ΔRH = 6.6%) within 120 ms and outputting 0.4~0.5 (ΔI/I_0_) signal amplitudes. The distinct state-dependent responsiveness enables tailored applications: SPG monitors physiological motions (e.g., pulse waves and joint movements) via conformal skin attachment, while D-SPG integrated into masks quantifies respiratory intensity with 3× signal enhancement during exercise. This work pioneers a sustainable candidate for biodegradable flexible electronics, overcoming trade-off limitations between mechanical integrity, signal stability, and dual responsiveness in starch hydrogels through synergistic network design.

## 1. Introduction

In nature, organisms adapt to their environment by sensing stimuli such as light, force, and humidity to alter their morphology [1,2,3,4,5]. For instance, mimosa leaves close under mechanical stress [6,7,8], while pine cone scales open or close in response to humidity changes [9,10]. Inspired by these biological behaviors, flexible sensors show significant potential in healthcare monitoring, bio-interfaces, and wearable devices by detecting external stimuli and transmitting electrical signals [5,11,12,13,14,15,16,17,18,19]. Hydrogels serve as ideal sensing materials due to their tissue-like properties: the hydrophilic 3D networks offer tunable mechanical characteristics and exceptional biocompatibility, enabling devices to conform to complex anatomical surfaces (e.g., joints) while maintaining functionality during large deformations. Among these, force-/humidity-responsive hydrogel sensors attract particular attention due to the ubiquity of these stimuli and their environmental compatibility [20,21].

Inspired by natural biosensors, various natural polymer hydrogel sensors have been developed [22,23,24,25,26], such as Wei’s cellulose nanofiber/carbon nanotube humidity switch (0.8 s response/2 s recovery) [27], Yang’s polydopamine-modified MXene/bacterial cellulose humidity actuator (1.6 s response) [28], and Wang’s ultra-low hysteresis response starch strain sensor (86 ms hysteresis) [29]. While these sensors have achieved fast response times, they suffer from single-responsive functionality, preventing their reliable operation under complex environments. Starch, one of the most abundant natural polymers, similar to other natural polymers, exhibits abundant intra-/inter-molecular hydrogen bonds, endowing it with excellent water absorption, retention capabilities, and biocompatibility, with established applications in food processing and artificial sensors [30,31,32]. But the unique gelatinization behavior of starch distinguishes it from other natural polymers. Under high-temperature water bath conditions, starch particles can absorb water, swell more than 50 times, and crack, releasing hidden hydroxyl sites [30,33]. Their unique gel behavior endows the starch molecules with excellent humidity-sensitive properties and provides an opportunity for the design of multi-responsive hydrogels.

Unfortunately, the insufficient rigidity of starch’s molecular chains compromises mechanical performance, hindering its practical application. To address the poor mechanical properties, Dodda enhanced the starch composite hydrogel sensor modulus (elastic moduli 3.9–6.6 MPa) through blending with chitosan, cellulose, PVA, and PEDOT:PSS [31]. But the components in the composite hydrogel are extremely complex, masking the moisture-sensitive properties of starch. Additionally, the intrinsic insulation of starch leads to signal interference, while the addition of conductive fillers can easily cause phase separation, except in the chemical modification or addition of additional additives [18,34,35]. Therefore, developing starch-based sensors that simultaneously achieve high mechanical strength, stable signal transmission, and multimodal responsiveness remains a significant challenge.

To address the challenge, this study developed a novel physicochemical dual-crosslinking strategy to fabricate starch-PEDOT:PSS-glutaraldehyde (SPG/D-SPG) hydrogels, aiming at achieving stable mechanical performance, stable signal transmission, and multi-responsive characteristics (Figure 1a). This approach uniquely leverages starch’s gelatinization and hygroscopicity within a stabilized dual-network architecture. The dried (D-SPG) and swollen (SPG) hydrogels exhibit distinct stimulus-response behaviors characterized through electrical signal monitoring, enabling simultaneous high mechanical strength (~266 kPa compressive fracture stress), exceptional signal stability (negligible drift, 2.7%), rapid dual response (pressure: 0.10 s; humidity: 0.16 s), and tailored state-dependent functionality (SPG for pressure, D-SPG for humidity). Finally, this starch-based hydrogel resolves persistent trade-offs among mechanical strength, signal stability, and dual-mode responsiveness through molecularly engineered network synergy.

## 2. Results and Discussion

In this work, we employed a “one-spot” synthesis strategy to prepare the dual-responsive hydrogels. As shown in the schematic diagram (Figure 1b), the starch aqueous solution, PEDOT:PSS (PP) solution, and glutaraldehyde (GA) solution were mixed at specific volume ratios, followed by ultrasonication to form a homogeneous blue suspension. Then, the suspension was stirred in an 80 °C water bath for 30 min to gelatinize the starch (Appendix A). During heating, hydrogen bonds between starch molecules ruptured, causing granules to swell and release molecular chains. As helical structures unwound and chains extended, the turbid suspension transformed into a viscous sol of mobile starch molecules. Simultaneously, glutaraldehyde (GA) reacted with starch hydroxyl groups to form stable ether bonds within 10 min at this temperature. Any unreacted GA was effectively removed by volatilization in the heated aqueous environment. Following this, the sol was spin-coated onto a home-made spacer, 4 × 4 cm^2^, and rested at room temperature for 12 h. During film formation, liberated starch chains underwent physical entanglement with PP chains, while concurrently participating in chemical crosslinking with GA through exposed hydroxyl groups, ultimately yielding homogeneous D-SPG films (Figure 1c). Immersing D-SPG films in deionized water resulted in fully swollen SPG hydrogels (Figure 1d) with a swelling ratio of 740 ± 53%. Additionally, reversible morphological switching between states was achieved through controlled drying and swelling (Appendix A). Each morphology exhibited distinct stimulus-responsive behaviors: D-SPG films leveraged starch’s hygroscopicity to rapidly adsorb ambient moisture and deform, enabling high-sensitivity humidity monitoring (ΔRH = 6.6%); meanwhile, SPG hydrogel served as a pressure sensor, withstanding mechanical loads while generating stable, noise-free electromechanical signals. This integrated approach demonstrates the dual-responsive (mechanical/humidity) and high-sensitivity properties of starch-based hydrogels, highlighting their significant potential for flexible sensing applications.

To elucidate the performance enhancement mechanism, we systematically investigated interactions among components. Starch molecular chains possess abundant hydroxyl groups that form intra-/inter-molecular hydrogen bonds, conferring crystalline ordering and structural integrity (verified by XRD, Figure 2a), which manifests as limited swelling without dissolution at ambient temperature. The characteristic peaks at 15.3°, 17.3°, 22.5°, and 24.3° represent the B-type crystalline structure of starch. However, the absence of effective inter-granular interactions impedes film formation. Consequently, hydrothermal gelatinization (80 °C) disrupts crystalline lamellae by breaking hydrogen bonds, releasing disordered molecular chains (Figure 2a). This irreversible transition to an amorphous state exposes hydroxyl groups, enabling film formation and humidity sensitivity. Nevertheless, the inherent low conductivity of starch compromises stimulus-response sensitivity in hydrogels. Moreover, while gelatinized starch chains retain hydroxyl groups capable of forming dynamic hydrogen bond networks, the lack of anchoring points renders these networks structurally labile under cyclic loading, which leads to network relaxation and consequent signal drift (poor signal stability). Compounding these limitations, native starch gels exhibit low mechanical strength, and incorporating conductive particles often further reduces ductility. To address these challenges, the dual-network architecture synergistically enhances mechanical and electrical performance: (i) PEDOT:PSS chains entangle with gelatinized starch, forming a physically crosslinked network that dissipates energy via reversible chain slippage; and (ii) GA covalently bridges starch hydroxyls, creating rigid nodes that prevent permanent network disruption (Figure 2b). This combination yields a “flexible-yet-stable” structure: PEDOT:PSS ensures ductility, while GA maintains integrity under load. Electrically, covalent crosslinks immobilize PEDOT:PSS chains, reducing conductive pathway reorganization during cycling and achieving drift-free signals.

Fourier-transform infrared (FTIR) spectroscopy effectively elucidated component interactions within the starch (ST) matrix through characteristic peak intensity shifts (Figure 2c). Starch exhibits two primary functional signatures: (1) hydroxyl groups (–OH) on glucose rings, manifesting as a broad peak at 3000–3500 cm^−1^; and (2) ether linkages (C–O–C) within glucose rings and skeletal structures, displaying absorption bands between 700 and 1200 cm^−1^. Specifically, these include skeletal ether bond bending/stretching vibrations at 700–900 cm^−1^ and characteristic C–O absorptions (including glycosidic bonds) at 900–1200 cm^−1^. The chemical crosslinking between GA and starch occurs via etherification, where aldehyde groups react with starch hydroxyls to form covalent C–O–C bonds. This reaction was verified through comparative analysis of normalized FTIR spectra, in which ST/GA composites exhibited significantly enhanced C–O bond absorption at 1150 cm^−1^ and 990 cm^−1^, alongside attenuated –OH stretching intensity at ~3300 cm^−1^ relative to pure starch, confirming successful etherification and establishment of chemical crosslinking sites.

Comparative analysis of pure starch and ST/PP spectra revealed significantly attenuated O–H peak intensity at ~3300 cm^−1^ (Figure 2c), indicative of hydrogen bonding between starch hydroxyl groups and sulfonate groups (–SO_3_^−^) of PSS. This interfacial interaction promoted uniform dispersion of PEDOT:PSS within the starch matrix, facilitating formation of a homogeneous, physically crosslinked network. Notably, the FTIR spectrum of the SPG ternary system closely resembled pure starch due to two factors (Figure 2d): (1) PEDOT’s conjugated backbone exhibited minimal infrared absorption, rendering it spectrally silent; and (2) characteristic absorption bands of PSS (particularly S=O stretching at 1040–1000 cm^−1^ and S–O stretching at 675–610 cm^−1^) were obscured by overlapping with strong starch polysaccharide signatures (C–O–C/C–O vibrations at 900–1200 cm^−1^) [32].

The conductivity of PEDOT:PSS originates from delocalized electrons within PEDOT’s π-conjugated polythiophene backbone and stable charge carriers provided by sulfonate groups (–SO_3_^−^) in PSS. Raman spectroscopy probed potential conductivity alterations upon composite formation (Figure 2e) [36]. The characteristic peak at 1433 cm^−1^ corresponds to the C_α_=C_β_ symmetric stretching vibration of thiophene rings in PEDOT, a mode highly sensitive to the chemical environment and directly linked to charge transport behavior. Upon incorporation with starch, this peak exhibited only a minor shift (Δ = +2 cm^−1^ to 1435 cm^−1^), indicating negligible disruption to PEDOT:PSS’s conductive network. In contrast, GA introduction induced a substantial redshift (Δ = −11 cm^−1^ to 1422 cm^−1^), signifying significant electron density redistribution within PEDOT chains. This electronic restructuring enhances charge delocalization and interfacial stability, thereby improving overall conductivity and operational stability in the SPG composite.

To elucidate GA’s regulatory effects on the electrical signaling of ST and ST/PP systems, we conducted force–electrical coupling tests using the setup depicted in Figure 3a and Appendix A, evaluating sensing performance across hydrogels (pure ST, ST/PP, ST/GA, SPG). Pure starch hydrogel exhibited substantial sensitivity fluctuations (ΔI/I_0_ = 0.4–2.5) under loading and significant drift (~49.5%) during cyclic loading (Figure 3b inset), attributable to dynamic hydrogen bond instability within its network. While ST/PP demonstrated near fivefold sensitivity enhancement (12.0) under identical loading and voltage conditions, signal drift (~58.3%) and instability intensified, indicating persistent mobility in the hydrogen bond network between PP and ST and the lack of covalent anchoring points. Conversely, ST/GA showed markedly improved signal stability with minimal drift (~0.6%) while maintaining consistent sensitivity (ΔI/I_0_ ≈ 0.6) from insufficient conductive pathways. These results confirm that ether bonds formed between GA and starch function as effective electrical stabilization anchors. Benefiting from a dual-network architecture integrating physical entanglement and covalent crosslinking, the ST-PP-GA (SPG) hydrogel demonstrated high sensitivity (~2.3) and stable drift (~2.7%) under cyclic loading with rapid response relaxation (~0.1 s) in Figure 3c and Appendix A. This confirms that the dual-network architecture, combining PEDOT:PSS entanglement for conductivity and GA-derived covalent crosslinks for stability, effectively eliminates baseline drift while maintaining sensitivity. Unlike the dense and brittle structure of pure starch hydrogel, SPG developed an open-cell, sponge-like porous framework (Figure 3d) that endowed exceptional flexibility and elastic deformability (Appendix A), effectively buffering external forces while resisting large deformations, including twisting and stretching (Figure 3e). The pure starch hydrogel could not maintain a stable shape under continuous load testing. Compression tests confirmed SPG sustained pressures up to 266.9 kPa without fracture, which is nearly 22 times higher than the 12 kPa pressure of low-GA-content starch. Meanwhile, other mechanical properties of SPG hydrogel were also strengthened—compressive elastic modulus, 26 MPa (from Figure 3f slope); and toughness, 13.9 kJ/m^3^ (calculated from stress–strain curve area)—significantly outperforming fragile starch hydrogels (Figure 3f). The optimized SPG formulation (0.75g PEDOT:PSS, 4 wt% GA) balances conductivity and mechanical stability: lower PEDOT:PSS (<0.5 g) compromises sensitivity, while higher amounts (>1.0 g) induce phase separation. Similarly, GA < 4 wt% yields poor ability to withstand pressure, and >5 wt% sacrifices flexibility (Figure 3g,h). This synergy ensures robust performance without trade-offs. Finally, we achieved >2-fold sensitivity enhancement over baseline systems (Figure 3g,h).

The D-SPG film exhibits exceptional humidity sensitivity due to abundant hydroxyl groups on starch molecular chains. When placed on a palm, the D-SPG film rapidly absorbs moisture within 0.2 s, disrupting interchain hydrogen bonds and forming new hydrogen bonds with water molecules. This induces localized chain hydration (“swelling”) that propagates macroscopic deformation (Figure 4a). In situ FTIR analysis confirmed the moisture–bond interplay: prolonged exposure to humid air (over water cup) redshifted the O-H stretching peak from 3300 to 3285 cm^−1^ with enhanced intensity, signifying strengthened hydrogen bonding (Figure 4b). SEM revealed an ultra-dense microstructure of D-SPG film structurally similar to pure starch (Figure 4c), with a compact architecture that facilitates rapid stress propagation from localized hydration sites to the entire film, enabling deformation at ultra-low humidity thresholds (Figure 4d). When integrated into circuits as a sensing element, dimensional changes during humidity-induced expansion alter conductive pathways, generating measurable electrical signals (Appendix A). Remarkably, The D-SPG film demonstrates exceptional humidity responsiveness (ΔRH = 6.6%) with a rapid 0.12 s detection time, 0.47 s signal output (0.4–0.5 amplitude), and full 0.34 s recovery, completing the entire response cycle in 0.93 s (Figure 4e,f). This ultra-fast cycling, enabled by engineered hydrogen bond reconfiguration, permits real-time monitoring of dynamic environmental changes (Movie S2). Moreover, the SPG and D-SPG also demonstrate over 200 cycles of stability (Appendix A), and other performances are summarized in Appendix A.

To comprehensively demonstrate the sensing advantages of SPG and D-SPG, we designed multiple flexible device prototypes. The exceptional conformality of SPG sensors enables seamless skin adhesion for detecting subtle physiological deformations, exemplified by wrist-mounted pulse monitoring, leveraging its rapid response characteristics (Figure 5a). Furthermore, SPG hydrogel exhibits distinct strain-dependent signaling, by which compressive stimuli generate nearly fourfold stronger signals than tensile inputs (Figure 5b). This differential response facilitated real-time wrist motion tracking through strategically placed sensor arrays, such as turning left or right (Figure 5c). For respiratory monitoring, D-SPG was integrated into masks’ inner linings, capitalizing on its humidity sensitivity and biocompatibility to detect metabolic states via breath moisture intensity. When exposed to 25 °C and 40 °C water vapor (simulating resting and exercising respiration, respectively), D-SPG generated threefold stronger signals at elevated temperatures (Figure 5d). This dual-responsive platform, SPG for mechanical sensing and D-SPG for humidity detection, enables application-specific configurations, while the sustainable design paradigm leveraging low-cost, biodegradable starch materials significantly advances hydrogel-based flexible sensing technology.

## 3. Conclusions

This work establishes a state-switchable starch-based sensing platform (SPG/D-SPG) through synergistic covalent–physical crosslinking, where GA-derived ether bonds stabilize dynamic networks to eliminate signal drift while PEDOT:PSS entanglement enables dual conductive pathways. The porous SPG hydrogel achieves mechanical–electrical synergy, withstanding 300 kPa stress with stable sensitivity (~2.3) and rapid 0.1 s response for high-fidelity biomechanical monitoring (e.g., pulse waveform capture). Conversely, the ultra-dense D-SPG film leverages starch’s hygroscopicity to detect minute humidity changes (ΔRH = 6.6%) within 120 ms, outputting 0.4~0.5 signals for breathing profiling (3× intensity gain at 40 °C). This dual-network starch hydrogel exhibits significantly faster humidity response (~0.12 s) than previously reported cellulose/MXene systems (~1.6 s) [28], and superior mechanical robustness (~300 kPa) and signal stability compared to conventional starch composites (~50 kPa) [30]. By exploiting state-dependent modality specialization, SPG for pressure/strain sensing, and D-SPG for humidity tracking, this system overcomes traditional starch hydrogel limitations and offers an environmentally friendly (biodegradable), low-cost-effective paradigm (<$ 0.10 per sensor patch), and large-scale production possibility (aqueous, 80 °C) for next-generation flexible electronics.

## 4. Experimental Section

### 4.1. Materials

Soluble starch and glutaraldehyde aqueous solution (GA, 25.0 wt%) were purchased from Sigma-Aldrich (St. Louis, MO, USA). Poly(3,4-ethylenedioxythiophene):polystyrene sulfonate (PEDOT:PSS) aqueous solution (1.4 wt%) was supplied by Adamas-Beta (Shanghai Adamas Reagent Co., Ltd., Shanghai, China). The mass ratio of PEDOT to PSS was about 2.5:1. Ultrapure water (≈18.2 MΩ cm) was used throughout the experiments.

### 4.2. Preparation of Dried Starch-PEDOT:PSS-GA (D-SPG) and Starch-PEDOT:PSS-GA (SPG) Hydrogel

A 10.0 wt% starch suspension was prepared by homogenizing 10.0 g starch in 90.0 g ultrapure water. Separately, a 2.0 wt% GA solution was formulated by diluting 1.0 g of 25.0 wt% GA stock with 11.5 g ultrapure water. For standard synthesis, 5.0 g starch suspension, 0.25 g PEDOT:PSS solution, and 1.25 g of 2.0 wt% GA solution were combined. The mixture underwent ultrasonication (Ultrasonic mode, sweep; Power, 40%; Frequency, 80 kHz) for 30 min to yield a homogeneous pre-crosslinked suspension designated as ST-PP(0.25)-GA(2%). This suspension was then stirred at 80 °C for 30 min to facilitate starch gelatinization, forming an SPG gel. Subsequently, 3.0 mL of SPG gel was cast into a custom 4 × 4 cm^2^ mold and air-dried at ambient temperature for 12 h to fabricate D-SPG films. Fully swollen SPG hydrogels were obtained by immersing D-SPG films in deionized water for 24 h. Variants with PEDOT:PSS quantities (0.25, 0.50, and 0.75 g) and GA concentrations (1.0~5.0 wt%) were synthesized analogously. Unless otherwise specified, subsequent experiments utilized hydrogels/films prepared with 0.75 g PEDOT:PSS and 4.0 wt% GA, yielding SPG hydrogels ≈2 mm thick and D-SPG films ≈ 15 μm thick. The SPG hydrogel required storage in low-temperature (4 °C) deionized water for its stable usage conditions, while D-SPG films were stored in a sealed bags to avoid moisture absorption in advance.

### 4.3. Calculation Method of Signal Sensitivity

D-SPG and SPG hydrogel films served as piezoresistive sensors in a 5 mV circuit. Current signals were measured for data acquisition, but, due to inherent resistance differences among the samples, the baseline (no-load) current varied. To standardize sensitivity measurements, we established the no-load current as *I*_0_ and the real-time current as *I*. Signal sensitivity was determined using the relative change to ensure consistent and comparable results across all samples, as in Formula (1):(1)Signal sensitivity = I−I0I0 = ∆II0

Meanwhile, the percentage drift was calculated as in Formulas (2)–(4):(2)I¯ = 1n∑i=1n∆II0i
where $\bar{I}$ is the average signal sensitivity of *n* stimuli throughout the entire experimental process, *i* is the *i*-th experiment number, and then(3)Drifti%=∆II0i−I¯×100
where ${Drift}_{i}(\%)$ is the signal drift of the *i*-th stimulus, and then(4)$$Drift=\;\frac{1}{n}\sum_{i=1}^{n}{Drift}_{i}$$
where *Drift* is the signal drift of the samples.

### 4.4. Calculation Method of Swelling/Deswelling Ratio

We quantified the swelling ratio (SR) of D-SPG films in deionized water at 25 °C, defined as follows:(5)$$SR(\%)\;=\;\frac{m_{t}-m_{0}}{m_{0}}\;\times \;100$$
where *m_t_* is the mass at time t, and *m*_0_ is the dry mass of D-SPG. The hydrogel reached equilibrium swelling (SR = 740 ± 53%) within 1 h (Appendix A).

The deswelling ratio (D-SR) of SPG hydrogels in air at 70 °C is defined as follows:(6)$$D\text{-}SR(\%)\;=\;\frac{m_{0}\;-\;m_{T}}{m_{0}}\;\times \;100$$
where *m_T_* is the mass at time T, and *m*_0_ is the wet mass of SPG. The hydrogel reached equilibrium drying state (D-SR = 83 ± 2%) within 2 h (Appendix A).

### 4.5. Characterization

Hydrogel thickness was quantified using a digital micrometer (DHG-050, Darmstadt, Germany). Chemical structures were analyzed by Fourier-transform infrared spectroscopy (FTIR, Thermo Fisher Nicolet iS20, Waltham, MA, USA) in attenuated total reflectance (ATR) mode. X-ray diffraction (XRD) patterns were acquired on a Bruker D8 Advance diffractometer (Saarbrücken, Germany) with Cu-Kα radiation (λ = 1.54 Å) at a scanning rate of 10°/min. Cross-sectional morphologies of freeze-dried, gold-sputtered samples were examined by field-emission scanning electron microscopy (FE-SEM, ZEISS SIGMA 300, Oberkochen, Germany). Raman spectra were recorded on an XploRA Plus 600 spectrometer (HORIBA, Paris, France) using 638 nm excitation. Mechanical properties were evaluated via uniaxial compression/tension tests on a universal testing machine (ZQ-990LA, Dongguan, China) at a 5 mm/min strain rate. Electromechanical sensing performance was characterized using a custom system comprising the following: (1) a stepper motor-driven compression stage, and (2) a signal acquisition unit with a Femto/Picoammeter (TH2691, Tonghui, Changzhou, China) and digital multimeter (DMM6500, Tektronix, Beaverton, OR, USA) under 5 mV bias. The sample size for mechanical and electromechanical measurements typically consisted of *n* = 5 replicates per condition to ensure statistical reliability. Ambient relative humidity (RH) was monitored with a digital hygrometer (UT333S, Dongguan, China).

## Figures and Tables

**Figure 1 sensors-25-05006-f001:**
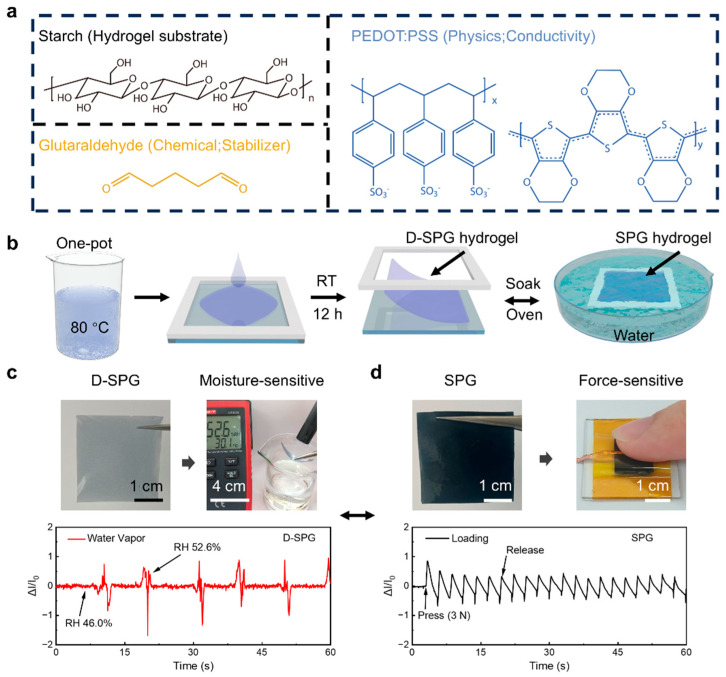
Dual-responsive starch-based hydrogel sensors. (**a**) Component materials and (**b**) schematics of fabrication processes of D-SPG and SPG hydrogels. (**c**,**d**) Photographs and stimulus signal curves of humidity-sensitive D-SPG and force-sensitive SPG.

**Figure 2 sensors-25-05006-f002:**
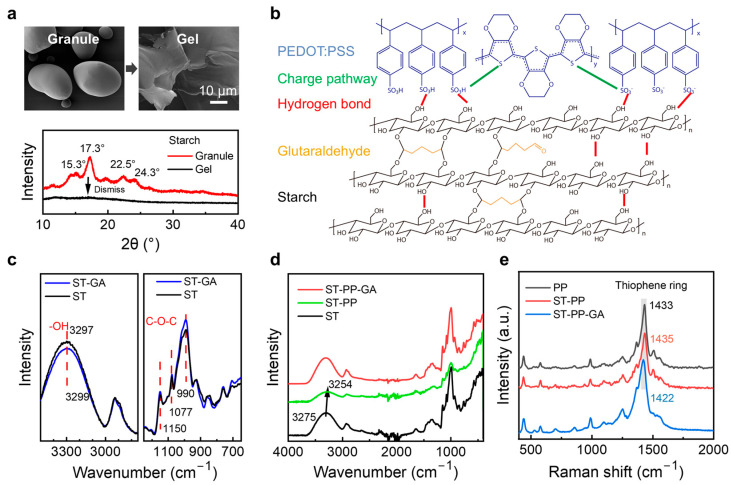
Characterization. (**a**) SEM and XRD images of starch granules before and after gelatinization. (**b**) Schematic diagram of molecular structure and intermolecular forces. (**c**,**d**) Fourier-transform infrared spectra of starch (ST), starch-GA (ST-GA), starch-PEDOT:PSS (ST-PP), and starch-PEDOT:PSS-GA (ST-PP-GA). (**e**) Raman spectra of PEDOT:PSS (PP), starch-PEDOT:PSS (ST-PP), and starch-PEDOT:PSS-GA (ST-PP-GA).

**Figure 3 sensors-25-05006-f003:**
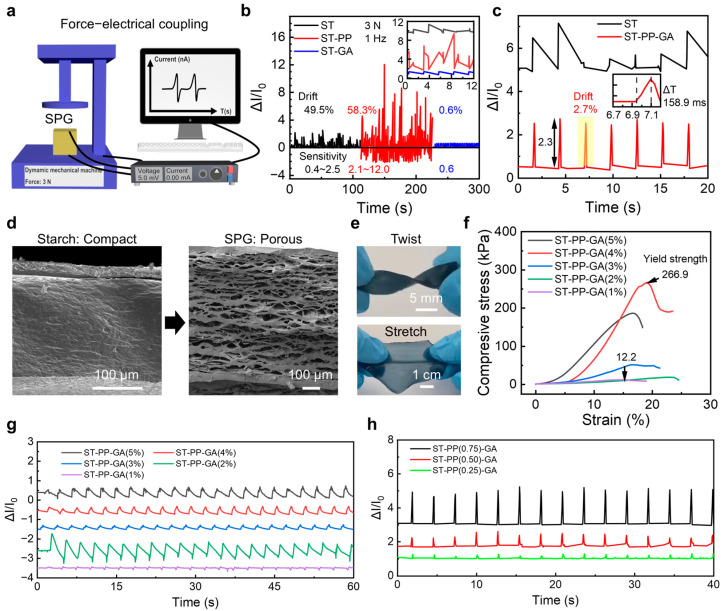
Force-sensing property of SPG sensors. (**a**) Schematic of the Force-electrical coupling device. (**b**,**c**) Signal sensitivity (ΔI/I_0_) of ST, ST-PP, ST-GA, and ST-PP-GA under the constant force of 3 N. All samples are the same size, 1 cm × 1 cm × 2 mm. (**d**) SEM images of starch and SPG hydrogel films. (**e**) Large deformations. (**f**) Compressive mechanical properties of hydrogel. (**g**,**h**) Signal sensitivity (ΔI/I_0_) of SPG sensors with various GA and PP contents under the constant force of 3 N. All samples are the same size, 1 cm × 1 cm × 2 mm.

**Figure 4 sensors-25-05006-f004:**
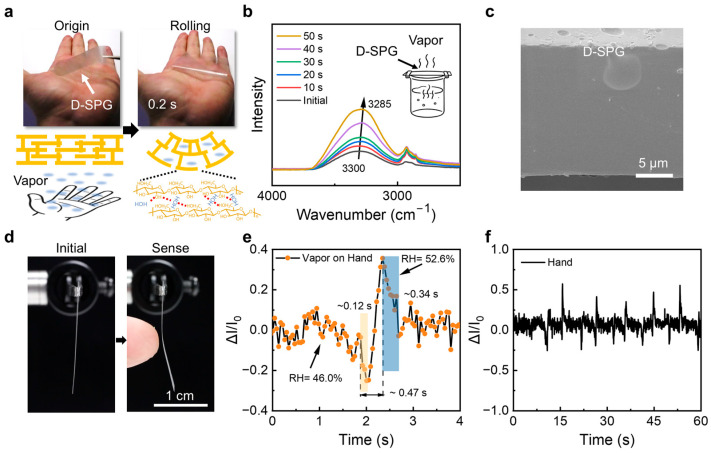
Humidity-sensing property of D-SPG sensors. (**a**) Photographs and schematic illustrations of the D-SPG film sensitivity to humidity. (**b**) Humidity-dependent ATR-FTIR spectra for D-SPG film as exposure to water vapor from 0 to 50 s. (**c**) SEM images of D-SPG films. (**d**) Photographs of D-SPG non-contact humidity-sensitive bending. (**e**,**f**) Signal sensitivity of D-SPG sensors under same humidity (54.6%). All samples are the same size, 2.0 cm × 0.5 cm × 15 μm.

**Figure 5 sensors-25-05006-f005:**
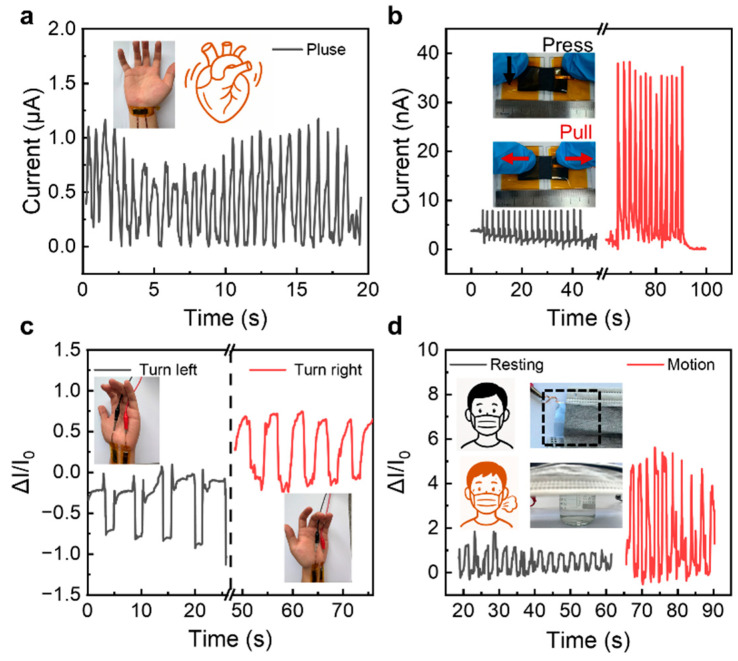
Demonstrations of flexible sensors with force/humidity stimuli. (**a**) Signal curve of SPG as a pulse patch. (**b**) Signal curve under compression and pulling stimuli of SPG. (**c**) Signal sensitivity curve of sensor arrays to track wrist motion. (**d**) Signal sensitivity curve of real-time monitoring of breath strength. Water vapor at temperatures of 25 °C and 40 °C simulate rest and exercise breathing, respectively.

## Data Availability

The data that support the findings of this study are available from the corresponding authors upon reasonable request.

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
