# Peer review of "Dual-Responsive Starch Hydrogels via Physicochemical Crosslinking for Wearable Pressure and Ultra-Sensitive Humidity Sensing"

_sensors, 2025, doi:10.3390/s25165006_

Round 1

Reviewer 1 Report

Comments and Suggestions for Authors

I read the manuscript “Dual-Responsive Starch Hydrogels via Physicochemical Crosslinking for Wearable Pressure and Ultrasensitive Humidity Sensing” and I would like to point out to several details:

1_) The abstract claims "high compressive strength (300 kPa)", but Fig. 3f shows this is the fracture stress, not necessarily the yield stress or elastic modulus. "Strength" often implies yield strength in materials science. The manuscript lacks key mechanical descriptors (modulus, yield point, toughness).

2)_ Figure Description – Clarity: "Fig. 1 Dual-responsive starch-based hydrogel sensors. a) The components and b) fabrication schematic diagram..."

Should be “Fig. 1. Dual-responsive starch-based hydrogel sensors: (a) Component materials, (b) schematic of fabrication process...".

3) Why: Ultrasonication parameters (power, frequency, time) are critical for reproducibility but omitted.

4) Figure 4 is missing from the text.

5) In Abstract and in Conclusion the authors say: ΔRH = 8% and in figure 1 it appears 46% and 52%. Please verify.

Reviewer 2 Report

Comments and Suggestions for Authors

This work introduces dual-responsive starch-based conductive hydrogel (SPG/D-SPG) developed using a straightforward "one-pot" synthesis strategy. This material integrates physical entanglement and covalent crosslinking to create a porous dual-network architecture, enabling it to function as a mechanically robust pressure sensor in its swollen state and an ultra-sensitive humidity sensor in its dried state. The developed hydrogels show promising performance for applications such as physiological motion monitoring and respiratory intensity quantification, offering a sustainable candidate for biodegradable flexible electronics. However, the work is not complete. It may be reconsidered after major revision.

Comments

  1. The detailed explanation of how the physical entanglement (PEDOT:PSS) and covalent crosslinking (glutaraldehyde) synergistically contribute to both mechanical strength and signal stability are not discussed.
  2. The paper claims to have eliminated signal drift. While Fig. 3b shows reduced drift for ST/GA and SPG compared to pure ST and ST/PP, a more quantitative analysis (e.g., percentage drift over time, or a metric for stability) are required.

  • A more detailed discussion or justification for selecting the specific optimal concentrations (0.75g PEDOT:PSS and 4wt% GA) in the main results section can be incorporated.

  1. The practical implications for scalability, ease of manufacturing beyond laboratory scale, or cost-effectiveness beyond "sustainable candidate" and "cost-effective paradigm" are not explored .

  1. Please use more concise phrasing or the use of active voice to improve readability.

  1. The tensile strength, elongation at break, and Young's modulus should be analyzed, especially since it's intended for wearable applications involving stretching and deformation
  • Please quantify the swelling ratio (e.g., mass increase over time) and swelling/deswelling kinetics in water are needed.
  • Evaluate the hydrogels' performance (mechanical, electrical, and sensing) over extended periods of time, multiple use cycles, and different environmental conditions (e.g., varying temperatures, humidity ranges, mechanical fatigue) for practical applications.
  1. The hydrogels are highlighted for their "exceptional biocompatibility"and are stated as "biocompatible" due to starch. However, the provided excerpts do not include any specific biocompatibility tests (e.g., cytotoxicity assays, in vivo studies).
  2. Thermogravimetric Analysis or Differential Scanning Calorimetry can be measured to quantify the water content in both SPG and D-SPG states.
  3. Detailed rheological measurements (e.g., storage and loss moduli) could further characterize the viscoelastic properties of the SPG hydrogel, especially relevant for understanding its conformability and behavior under dynamic stresses
  • While presenting a dual-responsive sensor, the cross-talk or interference between pressure and humidity sensing is not explicitly discussed. In complex environments, it would be important to understand if pressure changes can affect humidity readings or vice-versa, or how the sensor differentiates between stimuli.

Reviewer 3 Report

Comments and Suggestions for Authors

The manuscript presents an innovative dual-responsive starch hydrogel (SPG/D-SPG) combining physicochemical crosslinking for wearable pressure (300 kPa strength, 0.1 s response) and humidity sensing (ΔRH = 8%, 120 ms response), with applications in physiological monitoring and respiratory profiling. Key strengths include a robust dual-network design, comprehensive characterization (FTIR, SEM), and superior performance over existing starch composites.

In order to improve this article I have prepared additional proposals:
1. Elaborate on charge transport in PEDOTSS and humidity-induced microstructural changes; 2. Include long-term cyclic tests for mechanical/humidity stability;
3. Add cytotoxicity/skin irritation data per ISO 10993;  
4. Discuss environmental interference and energy requirements;
5. Improve figure labels (e.g., Fig. 3g-h) and define ΔI/Iâ‚€ in the abstract.

Reviewer 4 Report

Comments and Suggestions for Authors

Journal: Sensors

Title: Dual-Responsive Starch Hydrogels via Physicochemical Crosslinking for Wearable Pressure and Ultrasensitive Humidity Sensing

This study presents an innovative starch-based dual-responsive hydrogel with excellent pressure and humidity sensing capabilities, showcasing promising applications in wearable and biodegradable electronics. The integration of PEDOT:PSS and glutaraldehyde into a porous dual-network enhances both mechanical strength and electromechanical stability. The rapid, state-dependent response is particularly commendable. However, major revisions are needed to clarify the below concerns

  1. Under high-temperature water bath conditions, starch particles are capable of absorbing water and swelling by more than 50 times their original size, which can lead to cracking and the release of previously hidden hydroxyl groups. This unique gel behavior imparts excellent humidity-sensitive properties to starch molecules and offers potential for designing multi-responsive hydrogels. Based on this, it is suggested that starch molecules could function effectively as humidity sensors under elevated temperatures. However, it raises the question: how do the authors achieve responsive behavior at room temperature with such systems?
  2. The components within the composite hydrogel are highly complex, which may obscure the inherent moisture-sensitive properties of starch. Additionally, the insulating nature of starch can result in signal interference, and the incorporation of conductive fillers has the potential to induce phase separation. This phase separation could impact the uniformity and conductivity of the hydrogel, but the specific consequences and how they are mitigated are not fully clarified.
  3. The dried (D-SPG) and swollen (SPG) hydrogels demonstrate distinct stimulus-response behaviors, characterized via electrical signal monitoring. The study reports high mechanical strength (~300 kPa compressive), stable signals with negligible drift, rapid dual-responses (pressure: 0.10 seconds; humidity: 0.16 seconds), and functionality tailored to the hydrogel state—SPG for pressure sensing, D-SPG for humidity sensing. While the dual functionality has been well demonstrated, it would be beneficial to understand how these sensors perform under varying ambient temperatures and humidity levels. Additionally, what potential interference might occur at signal peaks, and what challenges could be faced during signal acquisition?
  4. Further clarification on the differences between D-SPG and SPG regarding transparency and electrical conductivity is requested, along with an explanation of how these differences influence their respective properties. Information about the stability of D-SPG and SPG upon immersion in deionized water over time—including swelling ratios and dimensional changes—would enhance understanding.
  5. How many cycles of water immersion and drying can these materials withstand without degradation? The authors’ comments on durability over multiple cycles, as well as their protocols for controlling the soaking process (e.g., specific pressures, volumes, durations), would be valuable. Additionally, insights into morphological and dimensional stability upon repeated recycling are requested.
  6. Regarding characterization, the SEM image in Figure 2a lacks a scale bar, which impairs reproducibility. The XRD pattern could be presented as a separate, clearly labeled figure rather than being merged with other data. Moreover, the absence of characteristic peaks in the gel state warrants an explanation—what factors contribute to this XRD observation?
  7. The choice of glutaraldehyde (GA) as a crosslinker is noted; it forms stable covalent bonds with hydroxyl groups in the hydrogel network. Clarification is needed as to why GA was selected over alternative crosslinkers and an assessment of its toxicity, especially considering potential biomedical applications.
  8. The authors state that the conductivity of PEDOT:PSS arises from delocalized electrons in the conjugated backbone and stable charge carriers from the sulfonate groups. Further insight into how PSS specifically contributes to electrical conductivity—beyond charge stabilization—would strengthen this explanation.
  9. Regarding the FTIR analysis, it is mentioned that the formation of ether linkages from GA crosslinking and sulfonic acid groups from PSS occur at concentrations below the detection limit, preventing observable spectral shifts. Could the authors specify the typical detection limits of FTIR and discuss how the actual concentrations of these linkages are estimated or confirmed?
  10. In the context of respiratory monitoring, D-SPG integrated into mask linings leverages its humidity sensitivity and biocompatibility to assess metabolic states via breath moisture. When exposed to water vapor at 25 °C and 40 °C—simulating resting and exercising respiration—the device produced signals that were approximately three times stronger at the higher temperature. How do the authors interpret this temperature-dependent response in relation to real-time breathing during rest and activity? Are these signals primarily influenced by humidity levels, temperature, or a combination of both? How can these contributions be deconvoluted to accurately interpret physiological states?
  11. What is the sample size used for the current response measurements? Additionally, how is the sensor thickness optimized for real-time applications? Providing a comparative analysis of sensors fabricated with and without GA crosslinking could offer valuable insights into the material's performance and the importance of crosslinker selection.
  12. Including a comparison table and schematic illustration detailing sensor response peaks and their correlation with stimuli would enhance the clarity and overall quality of the work.
  13. Including the above section in the article represents a constructive improvement; however, it would be beneficial to provide specific examples to support the authors' assertions. Furthermore, incorporating relevant citations would enhance the credibility of the manuscript (e.g., Gels 2025, 11(4), 220; Nanomaterials 2024, 14(3), 283; Chemical Engineering Journal, 502, 2024, 157752.).

Round 2

Reviewer 2 Report

Comments and Suggestions for Authors

Accept in present form 

Reviewer 3 Report

Comments and Suggestions for Authors

This version of the article can be published in the journal.

Reviewer 4 Report

Comments and Suggestions for Authors

The authors have addressed all reviewer comments, resulting in substantial improvements to the manuscript. The revisions align with the necessary standards of clarity, rigor, and scientific contribution. I recommend the manuscript for publication, as it now constitutes a valuable addition to the field, offering significant insights and advancements that are in line with the journal's objectives.